# Benefits and Biosafety of Use of 3D-Printing Technology for Titanium Biomedical Implants: A Pilot Study in the Rabbit Model

**DOI:** 10.3390/ijms22168480

**Published:** 2021-08-06

**Authors:** Sabrina Livia Ng, Subhabrata Das, Yen-Peng Ting, Raymond Chung Wen Wong, Nattharee Chanchareonsook

**Affiliations:** 1Department Oral and Maxillofacial Surgery, National Dental Centre Singapore, Singapore 168938, Singapore; cnattharee@yahoo.com; 2Department of Chemical and Biomolecular Engineering, National University of Singapore, Singapore 117585, Singapore; subhabrata@u.nus.edu (S.D.); chetyp@nus.edu.sg (Y.-P.T.); 3Department of Oral and Maxillofacial Surgery, Faculty of Dentistry, National University of Singapore, Singapore 119085, Singapore; denrwcw@nus.edu.sg

**Keywords:** milled titanium, direct metal laser sintered, titanium particles, bone implant, bone plate

## Abstract

Background: Titanium has been used in osteosynthesis for decades and its compatibility and safety is unquestioned. Studies have shown that there is release and collection of titanium in the organ systems with little note of toxicity. The gold standard is considered to be titanium osteosynthesis plate produced by milling methods. The use of customized titanium plates produced with 3D printing, specifically direct metal laser sintering, have found increasing use in recent years. It is unknown how much titanium is released in these printed titanium implants, which is known to be potentially porous, depending on the heat settings of the printer. We hypothesize that the amount of titanium released in printed titanium implants may be potentially more or equal compared to the gold standard, which is the implant produced by milling. Methods: We studied the biosafety of this technology and its products by measuring serum and organ titanium levels after implantation of 3D-printed versus traditionally fabrication titanium plates and screws in a pilot study using the rabbit model. A total of nine rabbits were used, with three each in the control, milled and printed titanium group. The animals were euthanized after six months. Serum and organs of the reticuloendothelial system were harvested, digested and assayed for titanium levels. Results: Organ and serum titanium levels were significantly higher in rabbit subjects implanted with titanium implants (milled and printed) compared to the control group. However, there was no significant difference in organ and serum titanium levels of subjects implanted with milled and traditionally fabricated titanium implants. Conclusions: The biosafety of use of 3D-printed titanium implants and traditionally fabricated titanium implants are comparable. With this in mind, 3D-printed custom implants can not only replace, but will very possibly surpass traditionally fabricated titanium implants in the mode and extent of use.

## 1. Introduction

The use of titanium in the form of biomedical implants (i.e., plates, meshes, and screws) has become very popular [1,2,3,4]. The success of this material lies mainly in its biocompatibility [5,6], biomechanical strength [7], and inertness to corrosion [8,9]. These implants are used mainly for reconstructive purposes and are traditionally shaped intra-operatively. With the introduction of 3D printing concepts in the 1990s [10,11], the natural progression from highly variable and intraoperatively time-consuming hand-shaped implants to pre-operatively prepared 3D-printed custom implant has become inevitable. Meanwhile, the biological compatibilities for use of this technology have not been fully explored [10,12]. To date, direct metal laser sintering (DMLS) has been the method chosen by medical device companies for production of their implants. There are other methods to 3D print titanium, for example, using fused deposition melting (FDM) technique of a titanium powder (Ti64) bound together with polymer into a filament and subsequently sintering the powder together in an oven. These methods, however, are not approved as a manufacturing method for surgical implants due to concerns about shrinkage of the implants during sintering, consistent reproduction and biocompatibility of the binder and post processing methods. Stock titanium implants are manufactured by milling blocks of titanium. There are cost differences between milled and DMLS produced implants, with milled costing more due to the fact that an entire block needs to be milled down into shape for the milled method; the DMLS method allows recycling of unused powder and also the printing of complex shapes. The safety of milled-stock-titanium implants is unquestionable, with titanium accepted as the gold standard for biocompatibility [6,13]. Three-dimensional-printed titanium rides on the back of traditional milled implants, as they share the same chemical structure. However, these 3D-printed titanium implants have microscopic structures highly dissimilar to those of traditional milled-titanium implants and have never been fully tested comprehensively for safety and weaknesses. This raises questions of whether the laser-sintering process is complete, whether it has any effect on the elution of titanium particles, and whether any discernible adverse local and systemic effect of leached titanium particles exists.

Titanium can enter the body from various sources (i.e., non-implanted titanium prostheses, food and other consumer products, as well as the environment). Titanium and its alloys are widely used in the forms of dental prostheses, such as orthodontic brackets, crowns, and dentures. Such prostheses, although not implanted in the body, can contribute to body titanium levels. In addition to its use in biomedical implants, titanium is widely used in the form of titanium dioxide (TiO_2_) for its whitening properties. In fact, over 95% of its global use is in this form. TiO_2_ is found in our food products; it is used to whiten, increase opacity, and modify texture [14]. As such, it can be found in foods such as confectioneries, white sauces and dressings, non-dairy creamers, and certain cheeses [15,16]. The mean diameter of food-grade TiO_2_ is approximately 200 nm, and the human daily intake of TiO_2_ is estimated to exceed 5.4 mg/day [17]. It has also been found that this consumption of TiO_2_ leads to negligible accumulation in the human body [18], and thus, it has been concluded that ingested TiO_2_ is excreted and found in wastewater-treatment plants [19]. As no publications have reported the removal of nano-sized titanium or TiO_2_ from wastewater-treatment plants [19], it is not unreasonable to assume that it will be released into the environment, where it may eventually be bioaccumulated in some of the food that humans consume.

There is evidence of the release of synthetic TiO_2_ nanoparticles from building-façade paints. These have been shown to contribute significantly to urban runoff after rainstorms [20]. It can also be found in sunscreens [16], and is widely used in dyes [21] and treatments for materials such as rubber [22]. Trace levels of TiO_2_ nanoparticles have been found in soil [23,24], water [23,25,26,27], sewage [28], and air [29,30].

In the oral and maxillofacial regions, milled and sintered titanium are expected to behave similarly in the body as they share the same chemical components of titanium 6-aluminium 4-vanadium alloy. Both milled and sintered titanium have been extensively used in the craniofacial region as surgical guides, implants for reconstruction of bony defects, and dental-prostheses retainers. They have also been approved for use as fracture-fixation devices, distraction-osteogenesis devices, and others.

### 1.1. Direct Metal Laser-Scintered (DMLS) Titanium Implants

The recent introduction of commercial 3D printing of custom titanium implants has enabled several new uses for titanium [10,31,32,33], from manufacturing custom cranial plates and screws, resulting in reduced operative times and patient morbidity, to printing porous-implant materials to encourage bone growth [34].

The manufacturing of DMLS titanium implants is conducted using one of the following processes:

#### 1.1.1. Pre-Formed Stock Titanium Implants

Such stock titanium implants are generally machined from wrought-titanium plates or rods, depending upon the geometry or size of the titanium-implant intended [35,36].

#### 1.1.2. Three-Dimensional-Printed Titanium Implants

Two types of additive-manufacturing techniques are available for rapid prototyping of titanium implants today––DMLS and electron-beam melting (EBM).

#### 1.1.3. Direct Metal Laser-Scintered (DMLS) Titanium Implants

This technique is an additive-manufacturing technique whereby an object is built incrementally layer-by-layer using powdered metal (in this case, titanium), a radiant heater, and a computer-controlled laser. A fixed layer of powder of accurate thickness, usually 0.1 mm, is placed and a high-powered laser is then directed and programmed to fuse the metal powder according to a computer-assisted-design (CAD) file. In this way, the object is built incrementally in layers until the process is complete [37,38,39,40,41,42].

#### 1.1.4. Electron-Beam Melting (EBM)

The main difference between EBM and DMLS is that EBM uses an electron beam to melt the titanium powder, while DMLS uses laser beams. The use of high-energy electron beams, thus, ensures complete melting of the powder particles and better bonding between them, resulting in a faster process with fully dense parts [43].

Both additive-manufacturing techniques use high-energy beams to melt titanium powder incrementally. Potential porosities can be included in such structures in the event of incomplete melting of the titanium powder. During the manufacturing process, each layer needs to be cooled prior to addition of the next. This repeated heating and cooling can potentially result in stresses in the metal superstructure, leading to a weaker structure than that of conventionally produced titanium implants. A study measuring the yield strength, ultimate tensile strength, Rockwell hardness, elongation, fatigue strength, and modulus of elasticity for EBM-processed Ti6Al4V compared to conventional cast-titanium implants showed better results for EBM implants [43]. Issues of porosity and weakness appear to affect DMLS more than EBM processes [37]. While this is so, the inherent porosity included in titanium structures manufactured using the DMLS technique can be advantageous with the new interest in open-pored morphology of implants, which may enhance osseointegration and vascularity [38,39,41,42].

Admittedly, the current literature is still undecided on the possible adverse effects these may incur [44,45]. However, evidence of additively manufactured implants causing elution of titanium particles from implants and resulting in local inflammatory responses has partially contributed to the number of implants removed post-operatively [46,47].

### 1.2. Leaching of Titanium Particles

Despite the benefits of DMLS over conventional milled-titanium implants, the former may leach more titanium particles into the body, as most printed titanium is combined from its powder form [48] using sintering as part of the laser and powder-bed-manufacturing technique. The concern arises as the use of solid-state sintering, a technique in which consolidation occurs below the melting point of a material, can result in diffusion of surface atoms, creating necks between adjacent powder particles that grow with time. These necks may significantly weaken the implant structure and result in microscopic disintegration of the implants over time, with concomitant release of titanium particles. It has, therefore, become necessary to test the difference in the release of particulate titanium from DMLS implants, in comparison to milled-titanium implants. There has been evidence of in vitro titanium-particle release from the implants to distant organs and lymph nodes [49], animal [50,51], and human studies [52,53,54,55]. The main method by which titanium nanoparticles are released from titanium implants and prostheses in the body is through corrosion of the implant’s surface. There are four main types of corrosion that can occur: (1) crevice corrosion, (2) pitting corrosion, (3) fretting corrosion, and (4) stress-induced cracking [56].

### 1.3. Adverse Effects of Released Titanium Particles

#### 1.3.1. Mutagenesis and Carcinogenicity Agents

Adverse effects found in animal studies were mutagenesis and carcinogenicity. Exposure of rats to TiO_2_ particles through inhalation of 250 mg/m^3^ of the particles showed that the rats developed lung adenomas and keratinizing squamous metaplasia, suggestive of squamous-cell carcinoma [57,58,59]. Other studies exposing rats to high concentrations of titanium nanoparticles resulted in increased incidence of malignant lung tumors [59,60]. There is currently no defined toxicity level for titanium.

The International Agency for Research on Cancer (IARC) has issued statements in 2006 and 2012 to the effect that, for humans, titanium dioxide should be classified as “possibly carcinogenic to humans,” as there is inadequate evidence in humans for the carcinogenicity of this material.

#### 1.3.2. Hypersensitivity Reaction

Hypersensitivity reactions to any metal must come from the body’s reaction to metal ions, following skin/mucosal contact, ingestion, or implant-corrosion processes [61]. According to Schramm and Pitto, titanium in its ionic form can bond with native proteins in the body to form haptenic antigens, which can in turn trigger the degranulation of mastocytes and basophiles and result in a type-I or type-IV hypersensitive reaction [62,63]. This reaction has been shown to lead to failure of titanium orthopedic implants.

Reports of titanium hypersensitivity date back to the early 1980s [64,65,66,67,68,69,70]. Titanium hypersensitivity is an exceedingly rare and little-known reaction to the previously considered bio-inert material. There has, however, been a mirage of reported non-specific titanium-hypersensitivity reactions.

### 1.4. Current Literature on Titanium Release from Implants

An in vitro study comparing the titanium released from various surface-treated, commercially pure titanium implants immersed in a simulated body-fluid solution showed that the maximum release of titanium was on the order of ~2.5 × 10^−3^ μg/cm^2^/h [71]. Based on rough calculations and assuming a linear scale, this would amount to a maximum accumulation of titanium on the order of 10.0 μg/cm^2^ at the end of 6 months. The same study also indicated a tapering graph for the release of titanium over time.

One group implanted vacuum-sintered titanium-felt implants into adult, male, white New Zealand Rabbits’ tibia. The serum and urine samples of titanium concentrations over a period of 12 months showed no significant differences among the control, sham, and test groups. Throughout the experiment, the mean serum-titanium concentrations were on the order of 5.0 × 10 ^−3^ μg/mL [72].

Woodman et al., in 1983, inserted milled tubular replacement segments into the femur, tibia, and humerus of female *Papio papio* baboons. They analyzed the titanium levels in these animal subjects’ lungs, kidneys, spleens, livers, surrounding muscles, and regional lymph nodes, and noted that the concentration of titanium found was related to both the surface area and implantation time. Elevated titanium concentrations were found in organs that served clearance functions, the lymph-reticular system (spleen, liver, lymph nodes), as well as in the lungs [73].

A cross-sectional clinical study analyzed the titanium-ion levels of patients who had undergone cementless total hip arthroplasty with a milled-titanium modular neck-femoral component, compared to the widely used non-modular CLS Spotorno system. The serum-titanium concentration detected 7–13 years after the implantation was on the order of 5.0 × 10^−3^ μg/m [74].

To date, there have been no studies comparing the amount of titanium released from conventional milled- and DMLS-titanium implants. Therefore, the objectives of this study are:To determine whether a difference exists between the amount of titanium particles released from milled and DMLS-titanium implantsTo compare the level of titanium release to that of the gold standard, the milled implants.

Since it is already well known that placement of titanium implants into the body causes detection of titanium in the organs (although decades long data of use shows little deleterious effects), our hypothesis was that as long as the level of release was not significantly higher compared to the gold standard, the use of printed titanium implants should be considered as safe as that of milled titanium implants.

## 2. Results

Two animals (from Test Group 2 and the control group) experienced premature death caused by hematoma of the neck following jugular-vein blood drawing. The protocol for blood drawing was reviewed, and subsequently, blood samples were withdrawn from the ear vein rather than the jugular with a smaller amount of blood taken.

Two control-group animals were sacrificed, and their organs were harvested on day 0. Five animals from the test groups continued to contribute to monthly blood sampling throughout the six-month post-operative period. There were no adverse post-operative signs, such as allergic reactions, wound infection, wound dehiscence, or osteolysis. The sutures at the surgical sites were removed, or had spontaneously exfoliated, within two weeks after the surgery.

### 2.1. Analysis of Titanium Concentration in Serum

#### 2.1.1. Comparison of Serum Titanium Levels Taken before Implantation (Month 0)

Using the paired *t* test, the average serum-titanium levels of the Control Group, Test Group 1 (milled implants), and Test Group 2 (DMLS implants) were 65.6, 76.0, and 65.8 µg/kg, respectively. There was no significant difference in serum-titanium levels in the test-group subjects compared to the control group.

Using the same statistical analysis, it was determined that there was no significant difference between the serum-titanium levels of the subjects in the milled and DMLS-implant groups.

#### 2.1.2. Trend of Serum-Titanium Levels in Test Group 1 (Milled-Titanium Implant) throughout the 6 Month Experimental Period

In Test Group 1 (milled-titanium implant), the average titanium concentration increased from month 0 to month 6 post-implantation—consecutively, these values were 76.0, 78.7, 79.4, 78.7, 83.9, 87.5, and 90.2 µg/kg.

Statistical paired *t* test analysis showed no significant difference between the serum-titanium levels in subjects before implantation and at the sacrifice point after 6 months had elapsed in the milled-titanium-implant group.

#### 2.1.3. Trend of Serum-Titanium Levels in Test Group 2 (DMLS-Titanium Implant) throughout the 6 Month Experimental Period

There was an appreciable and steady increase in serum-titanium levels across the implantation time period in Test Group 2 (DMLS-titanium implant); for months 0 to 6, these values were 65.8, 68.1, 72.6, 73.9, 75.0, 77.7, and 78.8 µg/kg, respectively.

The statistical paired *t* test analysis showed no significant difference between the serum-titanium levels in subjects before implantation and sacrifice in the DMLS-titanium implant group.

#### 2.1.4. Comparison of Average Serum-Titanium in Test Group 1 (Milled-Titanium Implant) and Test Group 2 (DMLS-Titanium Implant) throughout the Experiment

The average serum-titanium levels in both test groups were higher at the end of the experiment than at the beginning. There was an appreciable rise in both values throughout the experiment.

Using Scheffe’s method of statistical analysis, the mean serum-titanium level in Test Group 2 was significantly lower than that in Test Group 1 across 6 months of implantation.

At month 0, there was no significant difference between the serum-titanium levels between the test and control groups. The mean serum-titanium level of the DMLS-implant group was significantly lower than that of the conventional milled-implant group across the 6 months of implantation (Figure 1).

### 2.2. Analysis of Titanium Concentration in Organs

#### 2.2.1. Liver

The mean liver-titanium level of subjects in Test Group 1 was 422.4 µg/kg and that of Test Group 2 was 411.7 µg/kg. The mean liver-titanium level of subjects in the control group was 358.1 µg/kg.

Using ANOVA, it was found that the mean liver-titanium levels of the subjects in both test groups were higher than those in the control group. The mean levels in both test groups were similar, but those in Test Group 2 were slightly lower than those in Test Group 1.

#### 2.2.2. Spleen

The mean spleen-titanium levels were 419.9, 416.8 and 381.5 µg/kg for Test Groups 1 and 2 and the control group, respectively.

Again, using ANOVA, the mean spleen-titanium levels of both test groups were higher as compared to that in the control group.

There was no significant difference in mean spleen-titanium levels between the test groups.

#### 2.2.3. Thymus

The mean thymus-titanium levels in Test Groups 1 and 2 were 486.3 µg/kg and 487.0 µg/kg, respectively; it was 369.8 µg/kg for the control group.

Using ANOVA, the mean thymus-titanium levels of the subjects were found to be higher in the test groups than in the control group. There was no significant difference in the titanium level between the test groups.

There was no significant difference in the mean spleen, liver, and thymus-titanium levels between the test groups with DMLS and conventional milled implants after 6 months. The mean spleen, liver, and thymus-titanium levels in both test groups were significantly higher than those in the control group (Figure 2).

## 3. Discussion

The aim of the study was to compare the levels of titanium release in the reticuloendothelial system in rabbits implanted with milled with printed titanium implants in the form of bone plates. Our hypothesis was that if there were no significant differences in the levels between milled and printed titanium implants, we can conclude that printed titanium should be considered as safe as that of the gold standard milled implants.

Titanium plates were tested due to their common use in the body. The use of these implants in rabbit models followed the Institutional Animal Care and Use Committee (IACUC) guidelines for the smallest possible animal subjects available for use. The titanium plates were inserted into the rabbits’ femurs, as the size and location of the rabbit femur make it an ideal and accessible location for the implantation required in this study. A study duration of 6 months was chosen as the maximum release of titanium particles was expected to take place during the placement phase, where titanium powder is expected to be released from the implants due to friction with the implant drills during screw-hole preparation or due to friction with the securing screws during placement [75]. The transport of titanium powder would then be evident in the distant organs within 6 months. We expected the titanium within the implant to be stable thereafter, and thus, no further significant changes would occur after 6 months. This was based on various in vitro [76] and in vivo [77] studies for which the time point indicating maximum release of titanium appeared to be 3 months.

The ICP-MS system was chosen for the analysis of titanium particles, as it is a reliable and reproducible system capable of quantitatively analyzing trace titanium content [78,79,80,81]. Other similar systems exist, such as inductively coupled plasma-optical-emission spectrometry or inductively coupled plasma-atomic-emission spectroscopy; however, the ICP-MS system has been shown to provide lower detection limits for the measurement of trace elements and can eliminate scatter from interferences and contaminants [82].

Titanium, being a biologically inert material, is not expected to be metabolized by the body [83], and therefore, its accumulation in the organs indicates its release from titanium implants. An eluted titanium particle is expected to accumulate within regional lymph nodes [54]. The significance of the spleen is that it is the end point of the reticuloendothelial system. It is, therefore, assumed that the accumulation of titanium particles within the spleen indicates the presence of titanium particles in the lymph nodes. We tested the titanium content within the spleens of our subjects, as this organ could be more reliably accessed than regional lymph nodes. Previous studies [84] have demonstrated that the spleens of rabbits demonstrate the most significant bio-accumulation of titanium, with the greatest ratio of titanium content between the experimental groups and controls and the lowest standard deviation. The use of microwave digestion and analysis of samples with ICP-MS [50] allows for a quantitative comparison of titanium contents between samples. The IACUC guidelines state that the animal model used for any animal-research project should be “as small as reasonable” for such a study. This project involved placement of titanium plates and screws into the animal subjects. The rabbit model was, therefore, an animal model capable of accepting an implant of this size [85]. The benefit of using the rabbit model is that, considering its size, it is expected to experience several times the blood circulation, and hence, react with titanium implants, as compared to a human being over the same period of implantation. We, therefore, hypothesize that 6 months of implantation in our test subjects is equivalent to several years of titanium elution in an average human being.

Quantitative studies on the release of titanium from implants are limited, with no studies comparing the titanium released from milled- and DMLS-titanium implants. According to the results of a 12-month study by an orthopedic group from the University of Pennsylvania [72], 6.35 mm (diameter) by 2.3 mm (thickness) disc-DMLS titanium implants placed in the proximal-medial aspect of a rabbit’s tibia resulted in no further increase in serum-titanium content after 6 months as compared to the conventional milled-titanium implants. We, therefore, decided to conduct this pilot study for a period of 6 months for maximum impact with limited funds. An analysis of serum-titanium content also allowed us to determine the titanium-release trend in each animal group over the test period. This will allow us to observe the period of maximum titanium release, and hence, help us to plan the timeline for any further study.

It is of further importance to indicate that the initial serum-titanium levels of all our rabbit subjects, as well as the organ-titanium levels in our control-animal subjects, were higher than those reported by previous studies. The rabbit models used in this project were breeder rabbits. Unfortunately, the only rabbit models available at the time were from a commercial rabbit breeder in Singapore. Due to the greater age of these animal subjects compared to those used in previous studies, it is very possible that they were exposed to more TiO_2_ nanoparticles in the animal feedstock and accumulated more of these nanoparticles in their organs and serum.

There is no accepted reference level for titanium at an acceptable trace level in humans although there have been some studies conducted in blood and also some post-mortem studies. There are also no reference levels for titanium levels in animals or in the organ systems. Most studies agree on the level of titanium between the ranges of 50–150 µg/kg although in our study and for the reasons alluded to above, the levels are much higher even in the unoperated control group. We can only compare the levels of the printed implant group to that the gold standard milled implant group as a measure of safety [86,87,88].

In our study, placement of our titanium implant plates and screws inside the rabbits’ femurs did not require further bending for adaptation. However, the stock milled-titanium plates are typically bent and shaped prior to implantation when used in regular surgeries. This bending and shaping is not required for DMLS-titanium implants, which should have been shaped perfectly during the printing phase. This may have introduced a confounding factor in that the measured titanium levels in the conventional milled-implant group of our study may have been lower than the true titanium levels expected post-surgery. The friction between the drill bit and the titanium implant during preparation of the screw holes, as well as the screw with implant friction during implant placement, may contribute to the release of titanium particles from the implants. This was performed in our study for both implant groups to ensure that no additional confounding factors were introduced in the measurement of titanium levels in our subjects.

As with most fixation plates and screws, no significant wear was expected after implant placement and none was found in the study. We, however, expect more wear in load-bearing implants, such as in orthopedic hip and dental implants. Tribology, the study of wear and corrosion, is beyond the scope of this study.

The DMLS-titanium implants were fabricated from CAD Cam scans of the stock-milled titanium implants. Both implant dimensions were, therefore, identical. However, the DMLS-titanium implants were intentionally left unpolished, as can be observed through comparison between plates. This serves to increase the surface area of the implant in the body, and thus, the likelihood of titanium-particle elution. Further study of surface of plates from two different manufacturing mechanics will be analyzed using scanning electron microscopy, and the soft-tissue and bone responses of both titanium plates will be our subject of interest in later studies.

The serum-titanium levels indicated leaching of titanium particles from implants at a given time, whereas the organ-titanium levels indicated accumulation of titanium particles. Between these two parameters, it is easier and less invasive to study serum-titanium levels than organ-titanium levels, especially in the thymus and spleen, which are small and for which the entire organ is required for analysis. This study supports the hypothesis that there is no significant difference between titanium release into the body from DMLS and milled-titanium implants. There were significant differences in the levels of titanium particles found at distant sites in the body among the DLMS-titanium implant, conventional implant, and control group.

The size of the titanium particles released may affect their toxicity, which may be more detrimental than the absolute quantity of released titanium particles, as suggested by a rat study [89].

This study aimed to discover whether DMLS-titanium implants release more titanium particles than traditional milled-titanium implants. If titanium levels are shown to be significantly elevated in the DMLS group, this would indicate a need for future research in improving titanium-sintering technology to prevent the release of titanium particles. On the other hand, if DMLS implants are shown to be as safe as conventional milled implants, this would encourage more widespread use and the full benefits of DMLS implants to be reaped in various healthcare specialties.

As discussed earlier, the plates were not bent prior to implantation. Although bending of DMLS-titanium plates is not required prior to implantation, the milled-titanium plates are commonly bent and shaped for adaptation. In the future, this can be incorporated into the study design as the bending of plates may contribute to more titanium particles being released.

There is controversy in the literature in relation to the systemic distribution and accumulation of titanium released from implants. In our study, we opted to measure the titanium content from the lymph-reticular organs of our rabbits––spleen, liver, thymus. We chose these organs because of their higher likelihood of titanium-particle bio-accumulation. Furthermore, there have been human studies showing the presence of titanium particles in the liver and spleen in patients with titanium implants [90]. We did not analyze titanium content in the rabbit lungs, as another paper published by the same orthopedic group from the University of Pennsylvania [72] indicated that titanium has no effect on the rabbit-lung titanium levels. Instead, the lymph-reticular tissues were analyzed because of their high likelihood of showing an increase in titanium content.

This pilot study, involving nine animal subjects, was undertaken for the purposes of this research paper. The number of animals were admitted small for the reasons of this being pilot study: we were unsure if there would be any, or if there were significant differences in the levels of titanium detected. Thus, instead of using large numbers of rabbits unnecessarily, we elected to use a smaller sample size. The future study should include more animals for obtaining statistically conclusive results.

Sedation was achieved through IV-administered acepromazine at a dose 0.25 mg/kg. Emla cream was also applied to the puncture site prior to the venipuncture. Then, 8 mL of blood was drawn from the jugular veins of the rabbits for the first four blood draws. However, due to the mortality of the rabbits associated with this method, from the 5th month (6th blood draw) onward, this was later changed to 5 mL of blood drawn from the ear vein.

The use of the rabbit model, though effective in indicating the release of titanium particles from test implants, cannot provide a true representation of the same in the human body. The animal model closest to the human body will ultimately be the monkey. Should it be possible to repeat this study in monkeys, the results will be more representative.

Various animal studies have reported the bio-distribution of titanium particles accumulated in the liver, spleen, lungs, and kidneys [91]. These reports may, however, not be fully applicable in our study, as they usually involve intramuscular, intravenous, or intraperitoneal injection of titanium dioxide, or even oral administration of the material. As such, the distribution pattern may be dissimilar to what we are studying.

It may be useful and non-invasive to perform serial serum-titanium-level analyses in patients who undergo implantation with stock milled-titanium implants versus those who have DMLS-titanium implants placed for procedures such as fracture fixations and reconstruction. This would prove a non-invasive means of comparing titanium leached over time from implants between both groups.

As described previously, the size of titanium particles eluted may affect their toxicity within cells. Microparticles were found to be less toxic in rats than titanium nanoparticles [89]. Their impact and significance to human health should be determined from cell-line studies. Thus, a long-term study of titanium particles’ toxicity is recommended.

## 4. Materials and Methods

### 4.1. Titanium Implants

#### 4.1.1. Milled-Titanium Implant

A 4-hole straight mini plate of dimensions 22 mm (length) × 3 mm (width) × 1.25 mm (thickness) manufactured by Trinon^®^, Karlsruhe, Baden-Württemberg, Germany (the conformation most commonly used in the fixation of fractures), was employed as a conventional milled-titanium plate.

#### 4.1.2. Sintered (DMLS)-Titanium Implant

A plate with the same length, width, shape, and thickness was used (Figure 3). The sintered-titanium plate was designed via the CADCAM software program (AutoCAD, Autodesk, Portland, OR, USA) to mimic the design of stock implants. The sintered-titanium plates were printed via direct metal laser sintering by 4T Technologies (Singapore) using an electro-optical system selective laser-sintering printer, followed by rough polishing and tempering in an oven. The titanium-powder Grade 23 (Ti64 Eli extra-low interstitial, EOS, Germany) was used for the manufacture of the sintered (DMLS)-titanium implants.

Each implant was secured with two 1.5-mm titanium screws. Due to manufacturing limitations, the titanium screws used were commercially made, 6 mm milled titanium screws.

### 4.2. Experimental Animals

#### 4.2.1. Animal Group and Study Design

Nine female white New Zealand rabbits, weighing approximately 4–5 kg, aged 7 months to 1 year, were used in this study. All procedures were in accordance with the national guidelines for the care and use of laboratory animals, Singapore, and approved by the Institutional Animal Care and Use Committee (IACUC) of SingHealth (2016/SHS/1253). The research was conducted in the SingHealth Experimental Medicine Centre, accredited by the international Association for Assessment and Accreditation of Laboratory Animal Care (AAALAC). All three animals exhibited full adult dentition and were in healthy condition during the baseline evaluation. All animals were housed in individual cages. Solid food was provided to each animal daily and water was made available *ad libitum*. All animals were routinely checked for general health, body weight, and normal behavior.

The animals were divided equally into three groups—Test Group 1 received milled implants (*n* = 3), Test Group 2 received DMLS-titanium implants (*n* = 3), and the control group received no implants (*n* = 3). The subjects in the test groups had titanium plates placed in their bilateral femurs for a period of 6 months. The titanium levels were measured prior to surgery, at monthly intervals in rabbit serum, and at the end of the test period to test the rate of titanium elution into the animal system, while distant organs (spleen, thymus, and liver) were harvested at the end of the test period to examine the accumulation of titanium particles. The study design is shown as a study diagram in Figure 4.

#### 4.2.2. Implantation of Titanium Plates and Screws

The subjects partially fasted for 2 h prior to the surgery. Only drinking water was available during this period. Buprenorphine (0.01–0.05 mg/kg) was administered intramuscularly 30 min prior to the surgery, and then 6 h later as pain relief. The rabbits’ hind limbs were shaved prior to the surgery.

The surgery was performed under general anesthesia (GA). Anesthetic induction was performed using Ketamin (35–50 mg/kg) combined with Xylazine (5–10 mg/kg). Throughout the surgery, GA was maintained using 1–5% sevoflurane via a face mask. Vital signs and oxygen saturation were monitored throughout the surgery. The surgical site received a surgical scrub with povidone-iodine and 70% alcohol. Local infiltration of local anesthesia (2% Scandonest × 2.2 mL) was injected at the incision line. Under sterile conditions, a straight incision of approximately 3 cm was made along the middle part of the femoral bone. Sub-periosteal dissection was performed, and each titanium mini-plate, as described above, was secured to the rabbit’s femoral shaft with two titanium screws. Three animals in Test Group-1 were implanted with milled-titanium plates on each femur (×2 plates and ×4 screws, each animal). Three animals in Test Group-2 were implanted with sintered-titanium plates on each femur (×2 plates and ×4 screws, each animal). No operation was performed on the control group. A clinical picture of the plate and screws in place and a CT scan are shown in Figure 5. 

The surgical wounds were closed in layers using resorbable polyglactin 910 Suture 4-0 VICRYL RAPIDE™ (Ethicon Inc., Johnson and Johnson Company, Somerville, NJ, USA), after which a topical skin adhesive (Dermabond Mini, Ethicon Inc., Johnson and Johnson Company, Somerville, NJ, USA) was applied. The wound was then dressed in topical tetracycline cream.

Carprofen (1–2 mg/kg) was administered subcutaneously once a day for 3 days post-surgery as a means of pain relief. The antibiotic Enrofloxacin was titrated to 15 mg/kg body weight and administered orally twice a day for 6 days to prevent infection.

#### 4.2.3. Blood-Sample Collection for Serum-Titanium Analysis

In the test groups, animal subjects underwent sedation using intravenously (IV)-administered acepromazine at a dose 0.25 mg/kg for blood drawn (5 mL) on day 0 (prior to implant placement). This was repeated monthly for six months following plate implantation. Blood samples were allowed to clot for 30 min before the clot was removed using a centrifugation apparatus (KN70; Kubota, Tokyo, Japan) at 2200–2500 RPM for 10 min in a refrigerated centrifuge. The liquid serum was immediately transferred to a clean polypropylene tube and sent for analysis at the Department of Chemical and Biomolecular engineering, Faculty of Engineering, National University of Singapore (NUS). The serum yield was approximately 50% that of the blood drawn.

In the control animal subjects, only one blood sample was drawn; subsequently, the animals were sacrificed for harvesting their lymph-reticular organs for analysis.

#### 4.2.4. Sacrifice of Animals and Harvesting Lymph-Reticular Organ Tissues (for Organ-Titanium Analysis)

At the 6-month point, the animal subjects from Test Groups 1 and 2 were sacrificed. Their lymph-reticular organs (Figure 6), which included (a) spleen, (b) liver, and (c) thymus glands, were harvested at a weight of approximately 100–120 g and sent for analysis at laboratory of Chemical and Biomolecular Engineering, Faculty of Engineering, NUS.

### 4.3. Analysis of Serum and Organ Trace Metals in Biological Samples

The organs were weighed, processed via microwave digestion, and then analyzed using the ICP-MS system (ICP-MS system, Agilent 7500a^®^, Santa Clara, CA, USA), which can detect trace titanium elements. Each sample was analyzed thrice. In detail, a mixture of nitric and perchloric acids was used to decompose the tissue and serum-sample types.

Only plastic forceps, spatulas, or dropping pipettes were used to handle the tissue and serum samples. The tissue samples were removed from their shipping containers into a plastic high-density polyethylene dish for tearing prior to weighing. After recording the digestion-vessel weight, the balance was tared. The reading was ensured to be 0.0000 to ±0.0001 g. Then, 100 µL of 1000 µg/mL Yttrium internal standard was added to the digestion vessel. The weight was recorded, and the balance was tared. Between 0.5 and 1.5 g of tissue sample or between 0.1 and 0.15 g of NIST bovine-liver-QC sample or between 2.5 and 3.0 g of serum sample were added to the digestion vessel and the sample weight was recorded to the nearest 0.0001 g. In an acid fume hood, 3 mL of 70% nitric acid was added using a disposable LDPE-dropping pipette. Each group of digestions was accompanied by a blank and an NIST/SRM 1577b bovine-liver-QC sample. A group of samples comprised a full digestion block of samples. One group was equal to 24 vessels (22 samples and 2 controls). Since the NIST liver is dried, the sample weight was not expected to exceed 0.15 g. The digestion vessel was placed in the digestion block, which was maintained at 110 °C throughout the digestion. Brown nitrogen-dioxide fumes were observed within 5 min. The digestion was not left unattended for the first 15 min. The samples were completely dissolved within 15 min. The digestate was then swirled to render homogeneous. With the recommended sample weights, foaming was not a problem. Digestion of the sample was continued with nitric acid until the brown NOX fumes were barely visible. For safety reasons, an explosion shield was placed in front of the digestion block and the experimenter wore a face shield and heavy rubber gloves. Then, 2 mL of 72% perchloric acid was added to the mixture using a graduated 3 mL LDPE-dropping pipette.

The digestion continued at 110 °C for 16 h, and the digestate became a very pale yellow to water white and was allowed to cool to room temperature. The combined weight of the digestion vessel and digestate was recorded. The weight of the digestate alone was also recorded. The density of the digestate was determined to be 1.49 ± 0.02 g/mL. The volume of the digestate was obtained by dividing the digestate weight by its density, and then recorded. The final volume of the digestate was brought to 10 mL using 18-meg-ohm water. The volume of water added was also recorded. The weight of water added was calculated by multiplying the volume of water added by 0.997 g/mL and this too was recorded. The analytical balance was tared and the weight of the water was added (to the nearest ±0.02 g). The final sample solution was mixed after capping by hand-shaking. The sample was now ready for analysis.

The digestate was centrifuged at 10,000 rpm for 10 min and filtered (45 µm) to remove the suspended particles. The metal concentration in the digestate liquor collected after filtration was analyzed using ICP-MS (Agilent 7500a). Data were recorded as the mean of the three analyses performed on each sample, along with standard deviation and standard error.

### 4.4. Statistical Analysis

Three types of statistical analysis were employed in this study. A one-way analysis of variance (ANOVA) was used to analyze the organ-titanium levels (liver, spleen, and thymus) of all three groups at their respective sacrifice dates. A paired *t* test was used to analyze the pre-implantation and post-sacrifice serum-titanium levels of all three groups. An ad hoc comparison (Scheffé’s method of statistical analysis) was used to compare the linear regression of serum-titanium levels in the milled and DMLS-implant groups throughout the experiment. In all data analyses, a *p* < 0.05 was considered statistically significant, and the null hypothesis was rejected at this level of significance.

## 5. Conclusions

Within the limitations of this study, there is appreciable release and accumulation of titanium particles from both milled- and DMLS-titanium implants within 6 months following implantation. Both implants can, therefore, be considered comparably safe, based on the amount of titanium particles released.

Future studies may use different animals with more human anatomies and bone morphologies. A larger sample size will enable statistical analysis and performing serial serum analysis on the control subjects and sacrificing them at the same time as the test groups would allow for a better comparison among results. Furthermore, longer implantation duration should be considered until the elution of titanium particles becomes minimal. This can be determined in several ways, one of which would be to monitor serum-titanium levels until they become constant or begin to decline.

## Figures and Tables

**Figure 1 ijms-22-08480-f001:**
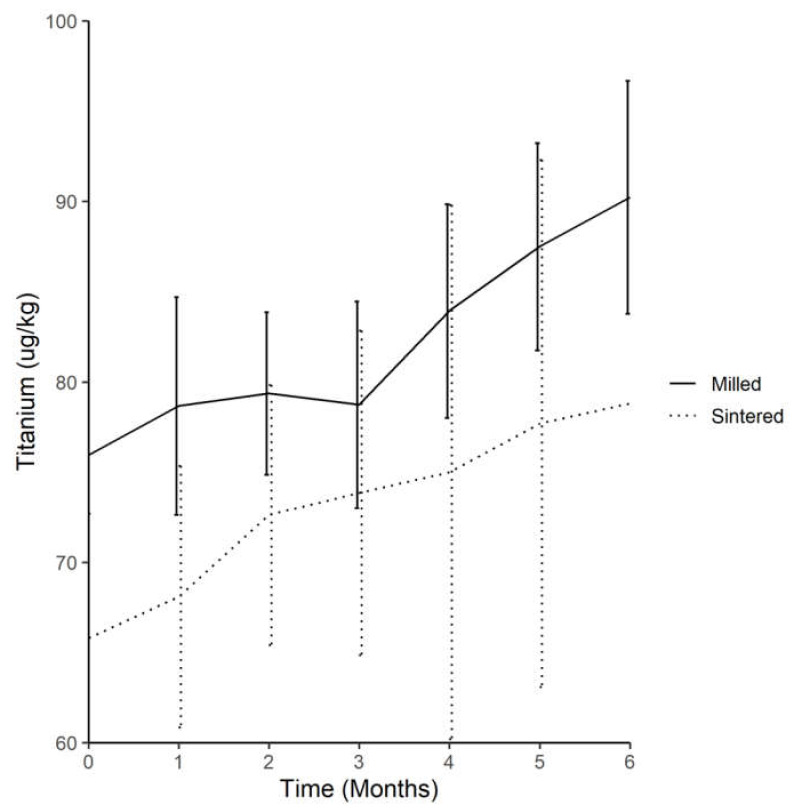
Comparison of the trend of the average serum titanium levels throughout experiment between both groups (Titanium (µg/kg)).

**Figure 2 ijms-22-08480-f002:**
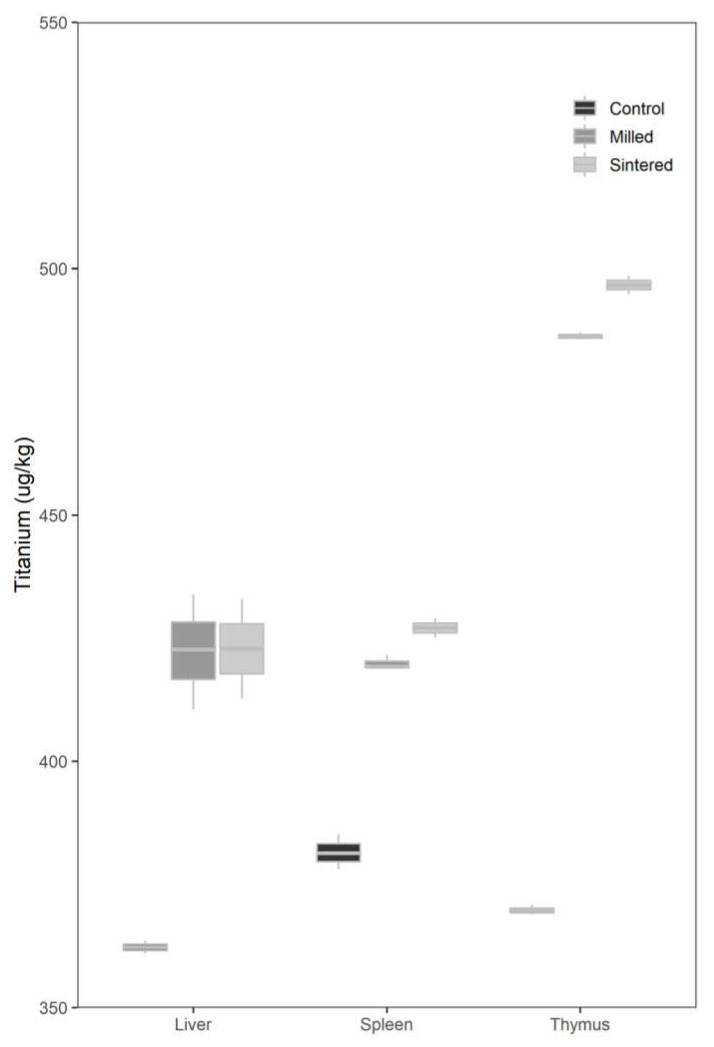
Box plot of organ titanium levels for the control and test groups.

**Figure 3 ijms-22-08480-f003:**
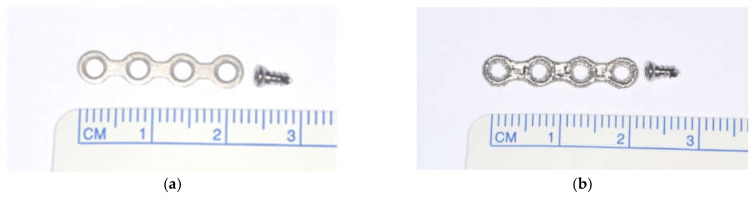
Titanium implant and screw use in the study: (**a**) milled titanium implant and (**b**) DMLS titanium implant.

**Figure 4 ijms-22-08480-f004:**
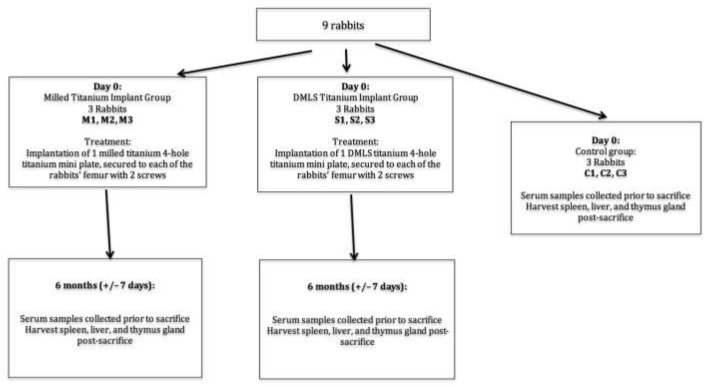
Schematic diagram of rabbit surgery.

**Figure 5 ijms-22-08480-f005:**
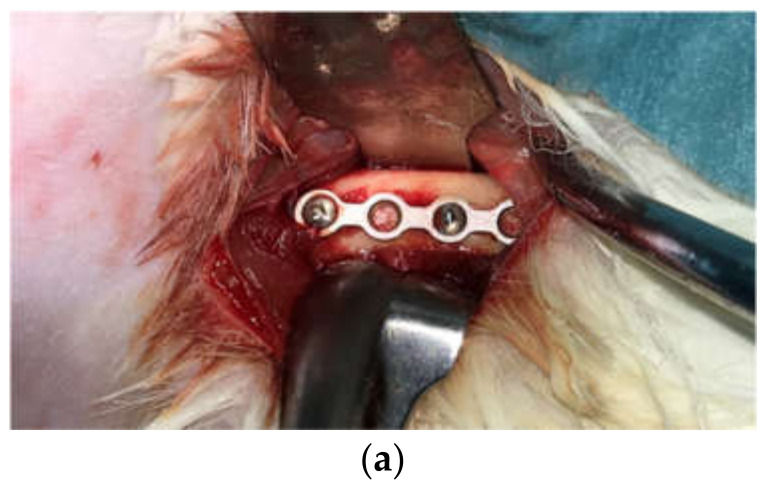
Titanium mini-plate inserted and secured with 2 titanium screws (**a**) a clinical picture, and (**b**,**c**) CT scan.

**Figure 6 ijms-22-08480-f006:**
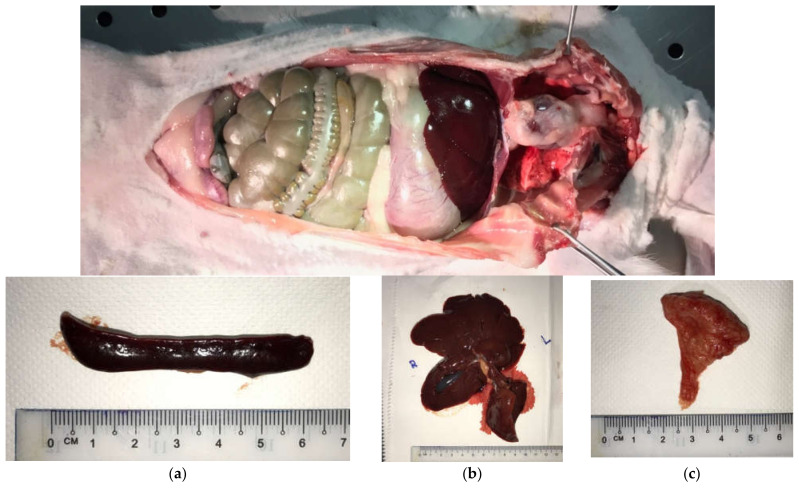
Rabbit abdomen exposed from the neck to the pubic region. The lympho-reticular of rabbit’s organs: (**a**) spleen, (**b**) liver and (**c**) thymus glands.

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
