# Peer review of "Benefits and Biosafety of Use of 3D-Printing Technology for Titanium Biomedical Implants: A Pilot Study in the Rabbit Model"

_ijms, 2021, doi:10.3390/ijms22168480_

Round 1
Reviewer 1 Report
This study determined whether a difference exists between the amount of titanium particles released from milled and DMLS-titanium implants. Additionally, the study determined whether titanium particles can be found in three organs in the body after implant placement. This study is interesting, has scientific value and should be published. However, the sample number seems very low. Therefore the merit of the results is low.
Introduction:
- Discuss the method of FDM 3D printing of an implant using Ti64 filament and then sinter refractory using 3D carbon in a Kiln (SAPPHIRE 3D – Sapphire3D and write the pros/cons of using the method.
- Discuss the economic (price of production/unit) advantage between milled and DMLS implants
Results:
- All Figure captions are vague. Need to search body text to understand the figure.
- Figures are not professionally presented
- Control missing for blood serum analysis [Figure 1]. What is M ave and S ave?
- Present Fig 2 and Fig 3 with error bar. A caption should be more descriptive. Add sample number in the error bar and Statistical significance level
Discussion:
- The manuscript needs to discuss the study results with other literature values. Ex. Compare the milled metal trace results with Rubio et al. 2008, Determination of metallic traces in kidneys, livers, lungs and spleens of rats with metallic implants after a long implantation time)
- Address the limitation of the study
Methods:
- What is the power of the same size? Discuss the low sample size.
- The animals were divided equally into three groups—Test Group 1 received milled implants (n = 3), Test Group 2 received DMLS-titanium implants (n = 3), and the control group received no implants (n = 3). However, two animals from control and test-2 died during the study. Does it mean n=2 for those groups? If so, the sample # is too low. Need to address this.
- This study is limited to metal traces only spleen, thymus, and liver tissue. The reason for not including kidney and lung for the study needs to be discussed
Reviewer 2 Report
The work evaluates 3D-Printing Technology for Titanium Biomedical Implants. The work is well structured, well written and interesting topic. The aim of the study is clearly stated. The background literature is described in sufficient detail. I suggest comparatives SEM pictures to be taken about the different samples and the histological analysis of the periimplant area.
Round 2
Reviewer 1 Report
The comments
- The title should add words "Pilot study" so that the reader knows that it is the first study on the subject matter.
- Write the sample size in the abstract.
- Add the gold standard value of metal trace acceptable and compare those values with the results found from the author's study.
